# *Penicillium citrinum* Provides Transkingdom Growth Benefits in Choy Sum (*Brassica rapa* var. *parachinensis*)

**DOI:** 10.3390/jof9040420

**Published:** 2023-03-29

**Authors:** Keyu Gu, Cheng-Yen Chen, Poonguzhali Selvaraj, Shruti Pavagadhi, Yoon Ting Yeap, Sanjay Swarup, Wenhui Zheng, Naweed I. Naqvi

**Affiliations:** 1School of Applied Science, Republic Polytechnic, Singapore 738964, Singapore; 2Temasek Life Sciences Laboratory, Singapore 117604, Singapore; 3Department of Biological Sciences, National University of Singapore, Singapore 117558, Singapore; 4NUS Environmental Research Institute, National University of Singapore, Singapore 117411, Singapore; 5Singapore Centre for Environmental Life Sciences Engineering, Singapore 637551, Singapore; 6College of Plant Protection, Fujian Agriculture and Forestry University, Fuzhou 350002, China

**Keywords:** Choy Sum, *Penicillium citrinum*, growth promotion, gibberellin, cytokinin, phytohormone

## Abstract

Soil-borne beneficial microbes establish symbioses with plant hosts and play key roles during growth and development therein. In this study, two fungal strains, FLP7 and B9, were isolated from the rhizosphere microbiome associated with Choy Sum (*Brassica rapa* var. *parachinensis*) and barley (*Hordeum vulgare*), respectively. Sequence analyses of the internal transcribed spacer and 18S ribosomal RNA genes combined with colony and conidial morphology identified FLP7 and B9 to be *Penicillium citrinum* strains/isolates. Plant–fungus interaction assays revealed that isolate B9 showed significant growth promotion effects in Choy Sum plants cultivated in normal soil, as well as under phosphate-limiting conditions. In comparison to the mock control, B9-inoculated plants showed a 34% increase in growth in aerial parts and an 85% upsurge in the fresh weight of roots when cultivated in sterilized soil. The dry biomass of such fungus-inoculated Choy Sum increased by 39% and 74% for the shoots and roots, respectively. Root colonization assays showed that *P. citrinum* associates directly with the root surface but does not enter or invade the root cortex of the inoculated Choy Sum plants. Preliminary results also indicated that *P. citrinum* can promote growth in Choy Sum via volatile metabolites too. Interestingly, we detected relatively higher amounts of gibberellins and cytokinins in axenic *P. citrinum* culture filtrates through liquid chromatography–mass spectrometry analyses. This could plausibly explain the overall growth induction in *P. citrinum*-inoculated Choy Sum plants. Furthermore, the phenotypic growth defects associated with the Arabidopsis *ga1* mutant could be chemically complemented by the exogenous application of *P. citrinum* culture filtrate, which also showed accumulation of fungus-derived active gibberellins. Our study underscores the importance of transkingdom beneficial effects of such mycobiome-assisted nutrient assimilation and beneficial fungus-derived phytohormone-like metabolites in the induction of robust growth in urban farmed crops.

## 1. Introduction

Cruciferous (Brassicaceae family) species represent a large number of economically important crops in global agriculture and food markets and constitute about 12% of the annual cultivation of leafy vegetables [1,2]. Brassicas are well recognized for being rich in antioxidants, carotenoids, polyphenols and more importantly for the presence of a unique group of anti-carcinogenic bioactive compounds termed glucosinolates. *Brassica rapa subsp. chinensis var parachinensis* (Choy Sum), a green leafy vegetable also known as Chinese flowering cabbage is a prolific and important cash crop in the tropics. It is widely cultivated and consumed in many Asian countries such as China, Japan and Singapore [2,3]. Recent studies on Choy Sum have reported the nutritional shifts of various metabolites depending on the growth stage and the effects of light intensity on the growth parameters and glucosinolate content [2,4,5]. However, the microbiome that shapes the rhizosphere environment of Choy Sum remains largely unexplored.

Plants harbor diverse microorganisms that engage in a continuum of interactions ranging from beneficial plant growth-promoting (PGP) to adverse pathogenic outcomes. PGP microbes play a key role in soil health and in the proper development of crop plants. Current trends in agriculture are focused towards reducing the input of inorganic fertilizers and the use of PGP microbes to achieve sustainability. Mentioned hitherto, the world of PGP microbes remains unexplored as far as Choy Sum is concerned, although some reports have described such beneficial microbes and endophytic fungi in other Brassicaceae species [6,7,8,9,10]. Plant growth-promoting rhizobacteria (PGPR) colonize roots and enhance plant growth directly and indirectly [11,12]. We aimed to explore the PGP mycobiome—fungi that are important components of soil microbiota and biomass—of Choy Sum. Rhizosphere- and root-associated fungi that are able to promote plant growth upon root colonization or through their secreted metabolites are functionally designated as plant growth-promoting fungi (PGPF) [13,14] and have gained significant attention in recent years. Over the decades, a growing list of PGPF from diverse genera including *Trichoderma*, *Penicillium*, *Phoma*, *Piriformospora*, *Fusarium*, *Aspergillus* and sterile non-sporulating fungi has been identified [15,16,17].

Previous reports showed that *Penicillium oxalicum* P4 and *Aspergillus niger* P85 isolates can produce seven and four organic acids, respectively, and can solubilise phosphate (Pi) and promote maize growth [18]. The fungal derived gibberellins and IAA by *Penicillium* sp. play a vital role in plant growth and development [19,20]. Similarly, in Arabidopsis, three fungal endophytes from water mint can increase the fresh and dry weight of Arabidopsis at 14 and 21 days post inoculation. Among them, *Phoma macrostoma* can increase both root area and depth at 21 days [21]. Active gibberellins (GAs) such as GA_1_, GA_3_, GA_4_, and GA_7_ were detected from pure cultures of *Phoma glomerata* and *Penicillium* species and they significantly increased the chlorophyll content in leaves and fresh and dry weight of shoots [22]. *Penicillium pinophilum* formed arbuscular mycorrhizae, which increase the plant dry weight, nitrogen content, P content and photosynthesis rate by 31%, 47%, 57% and 71%, respectively [23]. *Talaromyces pinophilus*, an endophytic fungus isolated from halophytic plants of Korea, can increase the plant height in comparison with the uninoculated wild-type [24].

Working on the hypothesis that the outcome of the host–PGPF interaction depends on the plant and microbial species, numerous fungi were isolated from the rhizosphere of Choy Sum and other plants and we describe here, for the first time, the association of a *Penicillium* species therein and its ability to promote plant growth. *Penicillium* species are known for their occurrences in diverse habitats and have received significant attention in the production of bioactive compounds. Investigations in recent years, including the research presented here, have reported *Penicillium* species as potent PGPFs that are capable of inducing plant growth by one or several direct or indirect mechanisms that include production of PGP phytohormones, solubilization of minerals, alleviation of drought and salinity stresses and antagonism against phytopathogens [20,25].

We report here two isolates of *Penicillium citrinum* B9 and FLP7 that showed promising growth phenotypes in Choy Sum in the laboratory as well as under soil-based greenhouse cultivation. Such induced plant growth was found to be due to nutrient acquisition enabled by the beneficial fungus and in part through fungus-derived and secreted phytohormones such as GAs and cytokinin(s). Our results demonstrate that symbiotic interactions with beneficial fungi play an important role in promoting plant growth and in sustainably increasing agricultural productivity.

## 2. Materials and Methods

### 2.1. Fungus Culture Conditions and Molecular Identification

The two fungal isolates, FLP7 and B9, used in this study are from our laboratory culture collection of beneficial fungal strains isolated from different environments, and possess plant growth promoting characteristics. The fungal strain FLP7 was isolated from the rhizosphere of Choy Sum grown under Pi-limiting conditions (12.5 µM), whereas B9 was from the roots of two-week-old barley seedlings grown in the soil. The GFP-labeled strain B9^GFP^ was further constructed in this study. The fungal strains were routinely cultivated in a prune juice agar (PA) medium [26] or in Potato Dextrose agar (PDA; VWR, Singapore) at 28 °C for 2 days in the dark and transferred to light at room temperature and grown for 7 days. Fungal cultures were grown in a complete medium (CM; 0.6% yeast extract, 0.6% casein hydrolysate and 1% sucrose) [27] for DNA extraction or for obtaining the culture filtrate or mycelial extract. Spore suspensions were prepared by rinsing the colony surface with sterile water, gently scraping the spores and hyphae off the medium followed by filtration through two layers of sterile Calbiochem Miracloth (Millipore, Burlington, MA, USA). The spore concentration was determined using a haemocytometer and adjusted with sterile water to the required concentration for plant assays.

To identify the fungal isolates, genomic DNA from mycelia grown in a PA medium was extracted using the MasterPure^TM^ Yeast DNA purification kit (Lucigen Corporation, Middleton, WI, USA) and used subsequently for PCR amplification using standard primers for the ITS or rDNA repeats. The PCR products were purified and sequenced. The sequences were used for NCBI BLAST analyses. The PCR primer pairs are listed in Appendix A. The PCR protocol used was 95 °C for 5 min; 35 cycles of 95 °C for 30 s, 48 °C for 30 s, and 72 °C for 1.5 min, 72 °C for 5 min; 95 °C for 5 min, 35 cycles of 95 °C for 30 s, 52 °C for 30 s, and 72 °C for 1.5 min, 72 °C for 5 min and 95 °C for 5 min, 35 cycles of 95 °C for 30 s, 60 °C for 30 s, and 72 °C for 30 s, 72 °C for 5 min, respectively.

### 2.2. Plant Growth-Promotion Assays

Seeds from Choy Sum or the Arabidopsis *ga1* mutant were surface sterilized and placed on a Murashige Skoog medium for germination. The Choy Sum seedlings were transplanted to autoclaved or non-autoclaved soil at 4 dpi (days post inoculation), respectively. The *ga1* mutant seedlings were transplanted to Phytatray II boxes containing the MS medium supplemented with cell-free culture filtrate from FLP7 or B9. The conidia were collected and diluted to a 1–5 × 10^5^ spores/mL suspension for inoculation. The fungus-inoculated plants were placed in a growth chamber for 2 days and then cultivated in a greenhouse until 21 dpi. The experiments were repeated three times each using 10–20 seedlings in each instance.

To assay for growth-promoting volatile metabolites, Choy Sum seeds were surface sterilized and grown on an MS medium as described above. The four-day-old seedlings in triplicate were transferred to Phytatray II boxes with the MS medium, together with fungal strains grown on prune agar, and the boxes were incubated at 25 °C, 70% relative humidity (RH) (day) and 23 °C, 50% RH (night) for 10 days. Barium hydroxide was added to the experimental set-up to quench excess carbon dioxide (CO_2_) in order to rule out its indirect beneficial effects in plant growth.

### 2.3. Agrobacterium-Mediated Transformation of B9

Agrobacterium-mediated transformation of target fungus was performed as described previously [28]. *Agrobacterium tumefacians* strain AGL1 carrying the appropriate Transfer-DNA vector/plasmid was grown at 28 °C in a LB medium containing 100 μg/mL Kanamycin overnight. The overnight AGL1 culture was diluted to OD_600_ = 0.15 using a standard induction medium [K salts (10 mMK_2_HPO_4_, 10 mM KH_2_PO_4_); M salts (2 mM MgSO_4_-7H_2_O, 2.5 mM NaCl, 4 mM NH_4_NO_3_, 0.7 mM CaCl_2_, 10 μM FeSO_4_); with glucose 5 mM, MES 40 mM, glycerol 0.5%, 100 μg/mL Kanamycin, 200 μg/mL acetosyringone] and incubated at 28 °C with gentle shaking at 160 rpm for 6 h. Simultaneously, conidia (fungal spores) were harvested from fully grown cultures of *P. citrinum* B9 strain (on a prune agar medium under light for about 1 week) and re-suspended to 1 × 10^6^/mL in distilled water. A sterile 0.45 μm nitrocellulose filter membrane was placed on an induction medium containing 200 μg/mL acetosyringone. A mixture of equal volume (100 μL each) of the AGL1 culture and the fungal conidial suspension was spotted and air-dried on the filter membrane. The plate was then incubated at 28 °C for 48 h. After the co-culture, all the growth on the filter membrane was scraped into 2 mL of sterile PBS (containing 200 μg/mL Cefotaxime, 60 mg/mL Streptomycin and 100 mg/mL Ampicillin) and was vortexed briefly. The resuspension in PBS was plated equally (200 μL) onto ten CM selection medium plates containing 200 μg/mL Cefotaxime (to kill *Agrobacteria*), 60 μg/mL Streptomycin, 100 μg/mL Ampicillin and 250 μg/mL Hygromycin. The selection plates were incubated at 28 °C until the transformed fungal colonies appeared (typically 3–5 days). The individual colonies were selected for mycelium preparation and DNA extraction as above. The primer pairs used for PCR amplification are listed in Appendix A.

### 2.4. Fungal Interaction/Colonization Assays in Choy Sum Roots

The Choy Sum seed germination and seedling preparation were as described above. The GFP-expressing B9 strain was prepared as above. The seedlings were submerged in the conidial suspension containing 1 × 10^4^ spores/mL. The interaction between Choy Sum and the cytosolic GFP-expressing B9 was analyzed using laser scanning confocal microscopy (Exciter, Zeiss) using the 10× water-immersion and 63× oil objectives. The excitation/emission wavelength (Ex/Em) was 488 nm/505–550 nm.

### 2.5. Plant Growth Assay under Low Pi Using P. citrinum Isolate(s)

The Choy Sum seeds were sterilized and germinated on a Murashige Skoog medium with low Pi (12.5 µM) for 4 days. The germinated seedlings were transplanted to autoclaved rice soil (with 0.11% (*w*/*w*) Pi) and grew for 21 days, as above. During the growth, no fertilizer was added. The average leaf area (cm^2^), root length (cm), dry weight of the roots and aerial parts (mg) were calculated for both control and B9- or FLP7-treated seedlings.

### 2.6. Extraction and Purification Methods for Detection of Phytohormones in P. citrinum

Certified standards of GAs (GA_1_, GA_4_, GA_20_) were purchased from OIChemim Ltd. (Olomouc, Czech Republic). Certified GA_3_ standard and formic acid were purchased from Sigma-Aldrich (Darmstadt, Germany). *Trans*-zeatin and *trans*-zeatin riboside standards were provided by Prakash Kumar (National University of Singapore, Singapore). Acetonitrile with 0.1% formic acid (Optima LC-MS grade) and methanol (Optima LC-MS grade) were obtained from Fluka Honeywell (Charlotte, NC, USA). Milli-Q water was used for preparation of the mobile phase (Millipore, Burlington, MA, USA). The Prime HLB SPE cartridge (200 mg, 6 cc) and Oasis MCX (200 mg, 6 cc) cartridge were supplied by Waters Corporation (Wilmslow, UK).

Standard stock solutions (1000 μg/mL) of GA_1_, GA_3_, GA_4_, GA_20_, trans-zeatin and trans-zeatin riboside were prepared individually in methanol and stored at −20 °C in the dark. Stock solutions were used to prepare working standard solutions for analytical experiments.

Extraction procedures for GA were adopted from the methods described previously [22,29,30]. Briefly, the liquid complete medium inoculated with *P. citrinum* isolates FLP7 or B9 was incubated at 28 °C at 180 rpm for 7 days. The purification of fungal samples was achieved via solid-phase extraction (SPE) with a reverse phase C-18 cartridge. Initially, fungal samples were extracted with ethyl acetate containing 1% (*v*/*v*) formic acid and centrifuged at 3000 rpm for 15 min. The supernatant of the mixture was then transferred and evaporated to dryness. For SPE purification, samples were reconstituted in 100% methanol and loaded onto the C-18 cartridge preconditioned with 6 mL methanol and then underwent equilibration with 6 mL distilled water. The cartridges were washed with 5 mL distilled water and the retained phytohormones (GAs) were eluted with 6 mL 100% methanol containing 1% (*v*/*v*) formic acid. The eluted extract was evaporated to dryness and reconstituted in 50% (*v*/*v*) methanol for the following LC-MS/MS analysis.

Extraction of cytokinins (trans-zeatin and trans-zeatin riboside) and subsequent sample clean-up and purification were performed using methods adapted from Morrison et al. (2015) [31]. Briefly, cell-free filtrates were snap frozen, lyophilized and subsequently homogenized in a cold (−20 °C)-modified Bieleski extraction buffer (Methanol/Water/Formic Acid, 15/4/1). Samples were allowed to extract passively, twice at −20 °C, and pooled supernatants were dried in a speed vacuum concentrator at ambient temperature (UVS400, Thermo Fisher Scientific, Waltham, MA, USA). Dried supernatant residues were reconstituted in 1 mL 1 M HCO_2_H and subjected to solid phase extraction on a mixed mode, reverse-phase, cation-exchange cartridge (Oasis MCX 6 cc; Waters, Wilmslow, UK). Trans-zeatin and trans-zeatin riboside were eluted with 0.35 M NH_4_OH in 60% CH_3_OH. Samples were evaporated and stored at −80 °C prior to analyses. Samples were reconstituted in initial mobile phase conditions (95:5 H_2_O:CH_3_OH with 0.08% acetic acid (CH_3_CO_2_H)) prior to analyses.

Headspace solid-phase microextraction (HS-SPME) sampling was used to collect volatile compounds emitted from *P. citrinum* [32]. The SPME fiber (50/30 DVB/CAR/PDMS, Agilent, USA) was exposed into the headspace of glass vials or Phytatrays II containing *P. citrinum* at 28 °C for 5–7 days in the dark. An uninoculated PA medium served as a negative/mock control for comparison. Determination of volatile profiles of *P. citrinum* was performed using untargeted gas chromatography coupled with electron impact ionization/time-of-flight mass spectrometry (GC-EI/TOF-MS) using Agilent 7890A platform following the manufacturer’s instructions.

### 2.7. Liquid Chromatography–Mass Spectrometry

LC–MS data were acquired on an Agilent 1290 Infinity coupled to an Agilent 6400 series Triple Quadrupole (Agilent, Santa Clara, CA, USA). An ultra-high performance liquid chromatography (UHPLC) system was integrated with Agilent 6490 controlled by MassHunter software B.06.00 (Agilent, Santa Clara, CA, USA).

For detection of GAs, 10 µL of extracts was chromatographed on a Zorbax RRHD SB-C18 (50 mm length × 2.1 mm diameter, 1.8 µm particle size) (Agilent, USA) with the column temperature set at 50 °C and auto-sampler temperature set at 4 °C. The mobile phase consisted of water acidified with 0.1% formic acid (Solvent A) and acetonitrile acidified with 0.1% formic acid (Solvent B). A gradient elution (flow rate 300 µL/min) consisting of 5% solvent B for 1 min followed by a linear gradient of 100% solvent B at 10.5 min was maintained until 13.4 min, followed by 5% solvent B at 13.5 min to 16.5 min for re-equilibration. Mass spectrometric detection was performed with a Triple Quadrupole in a negative mode with an Agilent Jet Stream ESI (G1958-65138) ion source using optimized monitoring reactions (Appendix A).

For detection of trans-zeatin and trans-zeatin riboside, 10 µL of extracts were chromatographed on a Kinetex C18 column (2.6 µm C18 100 Å, 100 × 2.1 mm) (Phenomenex, Torrance, CA, USA) with the column temperature set at 50 °C and auto-sampler temperature set at 4 °C. The mobile phase consisted of water acidified with 0.08% acetic acid (Solvent A) and methanol (Solvent B). A gradient elution (flow rate 300 µL/min) was used consisting of 5% of solvent B at 0 min followed 45% solvent B at 4 min, 75% B at 5 min followed by 95% B at 5.1 and was maintained until 6.1 min, followed by 5% solvent B at 6.2 min to 8.2 min for re-equilibration. Mass spectrometric detection was performed with a Triple Quadrupole in a positive mode with an Agilent Jet Stream ESI (G1958-65138) ion source using optimized monitoring conditions (Appendix A).

The mass spectrometer settings were as follows for all the phytohormones: source temperature 250 °C, gas flow 12 L/min, nebulizer gas pressure 35 psi, sheath gas temperature 350 °C and sheath gas flow 11 L/min. Data were recorded in the multiple reaction monitoring mode. All data collection, mass spectrometric and statistical analyses were carried out with the Mass Hunter Workstation software package: MH Acquisition B.05.00, MH Qualitative Analysis B.06.00. (Agilent Technologies, Santa Clara, CA, USA). All samples were randomized before LC-MS analyses.

### 2.8. Statistical Analysis

The data and comparison with controls were represented by using means with standard error. The significance of differences between the control and treatments was statistically evaluated using GraphPad (https://www.graphpad.com/quickcalcs/ttest1.cfm; accessed on 20 November 2022). Differences were considered significant at a probability level of *p* < 0.05 (*) or *p* < 0.01(***).

## 3. Results

### 3.1. Morphological Characteristics and Identification of Fungal Isolates B9 and FLP7

The morphological characteristics of the colonies and spores produced by B9 and FLP7 were analyzed by growing the isolates on a PA medium for seven days. The colonies were greyish green, moderately deep, with a raised center and with low or entire margins. On the PA medium, the conidiophores produced were mono-verticillate and sometimes bi-verticillate, with stipes smooth, phialides bottle-like, the conidia smooth walled, with globose to sub-globose and appearing in chains on the heads of phialides (Appendix A). The sequencing results of the internal transcribed spacer (ITS), large subunit (LSU) and small subunit (SSU) of 18S nuclear ribosomal RNA genes identified B9 and FLP7 to be *Penicillium citrinum.* As shown in Appendix A, the nucleotide sequence of the ITS region of B9 shared 100% nucleotide identity to the *P. citrinum*-type strain when searched for orthologous sequences in the NCBI Genbank database.

### 3.2. P. citrinum Improves Choy Sum Growth under Nutrient Rich Conditions

Standard plant–fungus interaction assays were conducted to check if *Penicillium citrinum* strains can induce or promote growth in green leafy vegetables. Inoculation of the spores of B9 to Choy Sum at the seedling stage could significantly increase the growth of plants both in sterilized soil as well as in non-autoclaved soil. Overall, the B9-inoculated plants were larger and grew taller than the uninoculated controls (Figure 1a–d). Compared with the mock control, the fresh and dry weight of aerial parts in the B9-inoculated plants increased by 34.8%, 39.5%, 41.2% and 25.4% in the sterilized or non-sterile soil, respectively (Figure 1e–h; *n* = 24, *p* < 0.05). Similarly, the fresh and dry weight of roots increased by 85.4%, 74.9%, 83% and 42.4% under the respective conditions (Figure 1e–h). These data helped us conclude that the B9 isolate of *P. citrinum* can significantly promote growth in Choy Sum both in the presence or absence of resident commensal microbiota in the rhizosphere. However, unlike B9, FLP7 did not promote growth in Choy Sum (Appendix A). The morphological traits of Choy Sum inoculated with FLP7 were comparable to the uninoculated controls. Furthermore, the fresh and dry weight of shoots and roots between FLP7 and its corresponding mock control did not show any significant differences (Appendix A) in nutrient rich conditions.

### 3.3. P. citrinum Improves Choy Sum Growth under Pi-Limiting Conditions

A previous report showed that the root endophyte *Colletotrichum tofieldiae* (*Ct*) promotes Arabidopsis growth under Pi-deficient conditions, though *Ct* did not display growth promotion under Pi-replete conditions [33]. Since the *P. citrinum* strain FLP7 was isolated from seedlings cultivated in a growth medium containing low levels of Pi, we tested whether the ability (if any) to promote growth in the host plants is restricted to Pi-limiting conditions. To further investigate this, sterilized soil with very low Pi content (0.11% (*w*/*w*)) was used to test this hypothesis. In comparison to mock treatment with sterile water, the Choy Sum seedlings grown in low Pi soil inoculated with FLP7 conidia showed a significant increase in overall growth and size of the plants. The average leaf area, root length, dry weight of the roots and shoots were 5.7, 2.0, 3.3 and 3.9 times that of the control, respectively (Figure 2a). We infer that the FLP7 isolate of *P. citrinum* can indeed improve the overall growth of Choy Sum, likely via facilitating the availability and/or uptake of phosphate in the host under low Pi conditions. To further check if the isolate B9 could also perform the same function under nutrient-limiting conditions, we used MS media with limiting amounts of Pi and checked the growth of Choy Sum grown along with B9. *P. citrinum* (B9) was able to induce Choy Sum growth under Pi-limiting conditions (Figure 2b). Furthermore B9 was found to possess a significant phosphate-solubilising activity as it was able to solubilize Ca_3_(PO_4_)_2_ in vitro (Figure 2c). As quantified in the bar chart, such Pi bioavailability and consequent root growth was further enhanced in planta in the presence of additional tricalcium phosphate (Ca_3_(PO_4_)_2_) (Figure 2d). We infer that *P. citrinum* can indeed improve the overall growth, specifically the root development in Choy Sum, likely via facilitating the bioavailability and/or uptake of Pi in the host under low Pi conditions.

### 3.4. P. citrinum Isolates Enhance Choy Sum Growth via Volatile Secondary Metabolites

Previous studies demonstrated that some beneficial fungi could secrete volatile organic compounds/metabolites to trigger plant growth and development [34,35,36]. To investigate if *P. citrinum* isolates B9 and FLP7 can induce similar VOC-based growth stimulation, we co-incubated 4-day-old Choy Sum seedlings with B9 or FLP7 colonies in Phytatrays for 10 days. Barium hydroxide was added to the experimental set-up to quench excess CO_2_ in order to rule out its indirect beneficial effects on plant growth. The tests for such volatile compounds indicated that compared to the mock control (the prune agar medium only), the size of the seedlings co-cultivated in Phytatrays with FLP7 or B9 colonies was significantly larger (Figure 3a–c; *n* = 18, *p < 0.05*). The fresh and dry weight of the shoots and roots of seedlings incubated with FLP7 (or B9) was 1.46, 1.14, 2.22 and 2.28 times higher than that of the respective mock controls (Figure 3c–f). Furthermore, the fresh and dry weight of shoots and roots of the seedlings incubated with B9 was 1.98, 1.63, 2.28 and 2.35 times than that of the uninoculated control plants (Figure 3c–f). Our preliminary analysis using solid-phase microextraction coupled with GC-MS identified seven differentially produced VOCs (Appendix A). The chemically identified compounds with individual retention time and the standards used for comparison (with CAS number) are mentioned. These results confirmed that *P. citrinum* secretes/exudes several putative volatile compound(s) which are likely responsible for indirectly imparting such beneficial effects on overall plant growth and development in Choy Sum.

### 3.5. Analysing the Colonization of Plant Roots by P. citrinum Isolate B9

To further investigate the mode of interaction of *P. citrinum* in the rhizosphere of plants, isolate B9 that showed promising effects in the conducted assays was transformed with the gene expressing a cytosolic enhanced green fluorescent protein (GFP). The resultant transformants were verified using PCR and sequencing, and the confirmed cytosolic GFP-expressing B9 strain tested for growth and conidiation (Figure 4a–c) was used for Choy Sum root inoculation assays. The host-associated hyphal growth of eGFP-expressing B9 was visualized at 12 h after inoculation.

However, no intracellular invasion or colonization within the roots was evident, even at 3 days post inoculation (Figure 5a). Similarly, the eGFP-tagged B9 strain was used to co-incubate with the roots of 4-day-old Choy Sum seedlings. Confocal microscopy revealed that the hyphae of B9 strain of *P. citrinum* also make contact with and adhere to the surface of Choy Sum roots (Figure 5b) but do not enter or colonize the root epidermal cells per se. These results demonstrated that the *P. citrinum* imparts beneficial effects via hyphopodial surface attachment and biotrophic interactions with Choy Sum roots.

### 3.6. P. citrinum Produces the Phytohormones Gibberellin and Cytokinin

Given the strong growth-enhancing effect on the host plants, we decided to evaluate whether *P. citrinum* isolate B9 produces and secretes any growth-promoting secondary metabolites or phytohormones. Towards this end, the axenic culture filtrates of B9 and FLP7 were analyzed using liquid chromatography–mass spectrometry (LC-MS) together with the requisite standards for two major classes of growth-promoting plant hormones: gibberellins (GAs) and cytokinins. The phytohormone detection was performed using optimized reaction monitoring conditions for the requisite standards as detailed in Appendix A.

Analysis of the cell-free extracts of B9 showed the presence of bioactive GAs including GA_1_ and GA_3_ and the inactive GA_20_ while in FLP7; the inactive GA_20_ can alone be detected even in the absence of the host plants (Appendix A). However, the results were variable, possibly due to the low abundance and/or instability of GAs produced and secreted by the fungus. Furthermore, GA-related inter-conversions should be considered, since only selective GAs were monitored in this study. On the contrary, GAs were absent or undetectable in the mock control (uninoculated growth medium) or within the mycelial extracts, thus indicating that the GAs detected in the fungal culture filtrates were most likely secreted by *P. citrinum* and not sourced from the growth medium per se, thus suggesting that the fungal GAs are most likely secreted extracellular metabolites.

To determine if the GA-related compounds produced by *P. citrinum* are, indeed, functionally active, an Arabidopsis GA-deficient mutant, *ga1*, and its isogenic wild-type Col-0 accession were germinated on a growth medium lacking or containing the culture filtrate of *P. citrinum* B9 or FLP7 (Figure 6). After 9 days, the *ga1* mutant supplemented with FLP7 or B9 culture filtrates showed shoot elongation and early flowering, with FLP7 showing comparatively better results (Figure 6a). Nevertheless, the wild-type Arabidopsis plants treated with the cell-free exudate of *P. citrinum* isolates showed early flowering compared to the control plants grown in the medium lacking the fungal culture filtrate (Figure 6b). Paclobutrazol is a known inhibitor of GA biosynthesis in plants [37]. In line with this, during our analysis, GAs were undetectable in culture filtrates and/or the corresponding mycelia from B9 treated with 10 µM Paclobutrazol. Hence, we used Paclobutrazol to further confirm if the fungal-produced GAs play a role in plant growth promotion. Interestingly, Choy Sum plants were unable to produce the effect of the culture filtrate and unable to establish when the inhibitor of GAs was added to the media (Figure 6c). This result indicates that the biosynthesis of fungal GAs is blocked in Paclobutrazol-treated *P. citrinum*. In contrast, the control set without the GA inhibitor showed the presence of GAs and consequent growth induction in the culture filtrates of both the beneficial isolates of *P. citrinum*. Taken together, these results revealed that the beneficial *P. citrinum* isolates FLP7 and B9 indeed produce a minor, albeit functionally significant amount of functional GAs, which is sufficient in trans to rescue the growth defects in the *ga1* mutant of Arabidopsis.

Apart from GAs, we evaluated the presence of cytokinins in the cell-free culture filtrates of these two isolates (B9 and FLP7) as well as in media extracts. We detected two cytokinins: trans-zeatin and trans-zeatin ribosides in the culture filtrates of *P. citrinum* B9 and FLP7. Interestingly, B9 produced higher amounts of these cytokinins compared to their presence within FLP7 (Appendix A, upper and lower panels). Taken together, we infer that *P. citrinum* (B9 isolate) produces relatively higher amounts of the two cytokinins, which are secreted; in addition, it also generates active GA derivatives, which are together likely responsible for the cross-kingdom increase in growth in B9- and/or FLP7-inoculated Choy Sum plants. However, detailed analytical studies are warranted to obtain further conclusive insights into the biogenesis and transport/uptake of *P. citrinum*-derived phytohormones or derivatives that are capable of inducing growth in the *Brassicaceae* host Choy Sum.

Overall, we conclude that the beneficial isolates of *P. citrinum* transiently associate with the host root surface, and physical entry or root colonization per se is likely not mandated or required for the observed beneficial effects in Choy Sum. The fungus-derived phytohormones, GAs and cytokinins likely play a key role in promoting robust growth and increased biomass in Choy Sum, an economically important urban vegetable crop in Singapore and Asia.

## 4. Discussion

The mode of action of rhizosphere fungi on plant growth promotion is complex. A wide range of factors interacting with multiple targets characterizes the response of beneficial fungi in plant growth and development. Our results demonstrate that *P. citrinum* is a plant growth-promoting fungus (PGPF) on the green leafy vegetable Choy Sum. *P. citrinum* IR-3-3 from the sand dune has previously been described as a beneficial fungal isolate in cereal plants [38]. More importantly, the analyses of IR-3-3 culture filtrate reports the presence of bioactive GAs, which displays growth-promoting activities on *Atriplex gemelinii.* The study of *P. citrinum* B9 and FLP7 is, from our perspective, an extended demonstration of a fungal species from different origins that leads to growth in different plant hosts. As an urban farm crop, Choy Sum is simple to grow in outdoor conditions. Improved shoot and root biomass are observed when Choy Sum plants are inoculated with *P. citrinum*, while no discernible symptoms during fungal colonization within plant roots are observed, indicating a biotrophic and beneficial interaction. To investigate the potential mechanisms of *P. citrinum* induced plant growth promotion, we evaluate the phytohormones in a culture filtrate. Intriguingly, the presence of GA_1_, GA_3_ and GA_20_ is detected either in the culture filtrate of B9 or FLP7. GAs are major plant growth stimulators that regulate cell elongation. They also play critical roles in seed germination, stem elongation and floral transition [39,40]. Complementation analyses reveals three lines of evidence for the bioactive function of *P. citrinum*-derived GAs in plant growth and development: (1) the rescue of the dwarf and flowering-defective phenotypes of *ga-1* mutant lines; (2) the upregulation of Arabidopsis plant size upon exogenous supply of fungal culture filtrate; and (3) the growth retardant by GA inhibitor to the Choy Sum inoculated with culture filtrates of B9 or FLP7. Thus, the growth-promoting effect of culture filtrate inoculation on plants highlights the bioactive GA production capacity of *P. citrinum*.

Our study adds to the emerging role and importance of mycobiota-derived phytohormones and/or derivatives that contribute to the functional aspects of growth benefits in the host plants. Many fungi have been shown to produce phytohormones such as auxin, jasmonates and/or GAs [19,22,30,41,42,43]. GAs play key roles in root colonization and are well known for their role in various developmental processes in plants, including germination and stem elongation [44]. Several studies have documented that production of IAA and GAs by *P. citrinum* and other *Penicillium* sp. were most effective in promoting plant growth [20,38]. Apart from GAs, the presence of cytokinins is also observed in the culture filtrate of *P. citrinum*, the other important phytohormones involved in maintaining cellular proliferation and differentiation. Zeatin production has been documented in *Piriformospora indica* and *T. harzianum* [45,46], and it has been shown that trans-zeatin cytokinin biosynthesis is crucial for *P. indica*-mediated growth stimulation in Arabidopsis [46]. This evidence suggests that *P. citrinum* could mediate plant growth and development at different stages by influencing the phytohormone balance in the host plants.

In addition to phytohormones, volatile secondary metabolites produced by rhizosphere microbes are one of the important chemical stimuli involved in plant growth promotion. In our study, we have shown the increased shoot and root biomass of Choy Sum, even in the absence of physical contact with *P. citrinum*. CO_2_ has been verified as a constituent of the plant growth-promoting volatiles produced by microorganisms in a hermetic system [47]. Interestingly, the enhanced plant biomass is not affected by the exclusion of CO_2_ in the hermetic system, suggesting a minor role for fungus-derived CO_2_ in plant growth promotion. Recent studies have identified the role of VOCs as the signaling molecules in the plant–fungus systems [34,48]. In general, VOCs are thought to be good candidates for below-ground communication due to their great diffusivity in soil [49]. Consistent with this, several fungal VOCs display growth-promoting effects on plants. For example, ectomycorrhizal fungus *Laccaria bicolor* produces sesquiterpenes (SQTs) as bioactive agents and promotes lateral root formation in Populus and Arabidopsis plants [50]. Our preliminary analysis using solid-phase microextraction coupled with GC-MS identified seven differentially produced volatile metabolites, including the sesquiterpene Longifolene-(V4) as one of the VOCs produced by *P. citrinum*. However, it remains to be confirmed whether the growth-promotion effect of *P. citrinum* in Choy Sum is caused by such volatile compound(s)/sesquiterpene. Interestingly, methyl salicylate (MeSA), which is a volatile plant defense compound produced in response to pathogen attacks [51] and plays phytohormone-like regulatory roles [52] was also detected in the *P. citrinum*-associated VOCs. MeSA is derived from SA by an SA carboxyl methyltransferase. A previous study confirmed that SA can regulate processes such as seed germination, vegetative growth and photosynthesis, in addition to its role as a regulatory signal mediating plant response to abiotic stresses such as drought and chilling [53]. Moreover, only a few compounds could be detected in the shared atmospheric space of *P. citrinum*. It is possible that most of these VOCs are produced in meagre amounts that are below the detection limits, and/or are labile and produced only during a specific developmental stage such as during direct physical interaction with plant roots. Newer methods could be utilized in the future for the capture and identification of such VOCs. Longer capture times such as 10–14 days could also be tested in future experiments. Lastly, much older, i.e., 2–3-week-old fungus could be harnessed for VOC collection to improve the quantity and quality of the emitted VOCs [54]. Although we have shown that VOCs produced by *P. citrinum* likely promote the host plant growth, their detailed mode(s) of action and ecological benefits have not been fully elucidated. Thus, further identification and functional characterization of growth-enhancing VOCs produced by *P. citrinum* will be necessary to fully understand such beneficial interactions in fungus–plant systems.

## 5. Conclusions

Under natural conditions, rhizosphere microbes have different interactions with the plant hosts, ranging from commensalism to mutualism. Reciprocally, plants can also shape the rhizosphere microbiome for their growth, development and abiotic and biotic stress tolerance. Although many studies have been conducted, the understanding of molecular mechanisms associated with the beneficial microbe and host interactions is still far from complete. Next-generation sequencing technology, combined with the development of metatranscriptomics, metaproteomics and metabolomics, will push forward the understanding between hosts and the rhizosphere microbes. In this study, we showed that beneficial *P. citrinum* isolates can promote growth in Choy Sum (an important urban crop for food and nutritional security) via nutrient assimilation, and via secreted phytohormone(s) and/or putative volatile compounds. We demonstrated that rhizosphere fungi can be considered as a useful bioresource, enhancing soil fertility and promoting sustainable plant growth. Integrating the knowledge of mycobiome community composition, beneficial microbial consortia, volatile signals and mutual interactions could aid in sustainable agriculture in urban settings. Future studies will be directed at understanding the physiology and mechanism-of-action of these fungal phytohormone(s) in such cross-kingdom growth promotion and resilience (if any) in other important food crops in traditional and urban agriculture.

## Figures and Tables

**Figure 1 jof-09-00420-f001:**
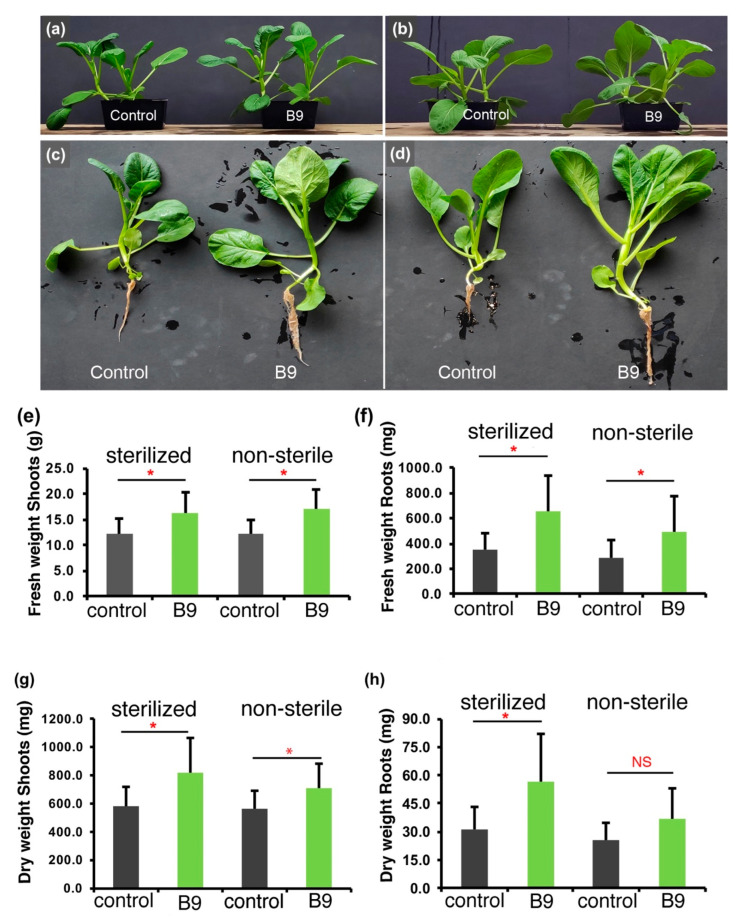
The morphological traits of Choy Sum inoculated with *P. citrinum* and the plant growth promotion effect in sterilized or non-sterilized soil (**b**,**d**). (**a**,**c**) The morphological traits of Choy Sum grown for 21 days in autoclaved soil (left, water; right, inoculated with B9 conidia); (**b**,**d**) The morphology and growth characteristics of Choy Sum plants grown in non-autoclaved soil for 21 days (left, water; right, inoculated B9 spores). (**e**–**h**) Bar charts showing quantification of the fresh and dry weight of shoots/aerial parts (**e**,**g**) and roots (**f**,**h**) under the two growth conditions, respectively. Data represents means ± SDs from 3 replicates consisting of 16 plants in each instance. Differences were considered significant at a probability level of *p* < 0.05 (*). NS: not significant.

**Figure 2 jof-09-00420-f002:**
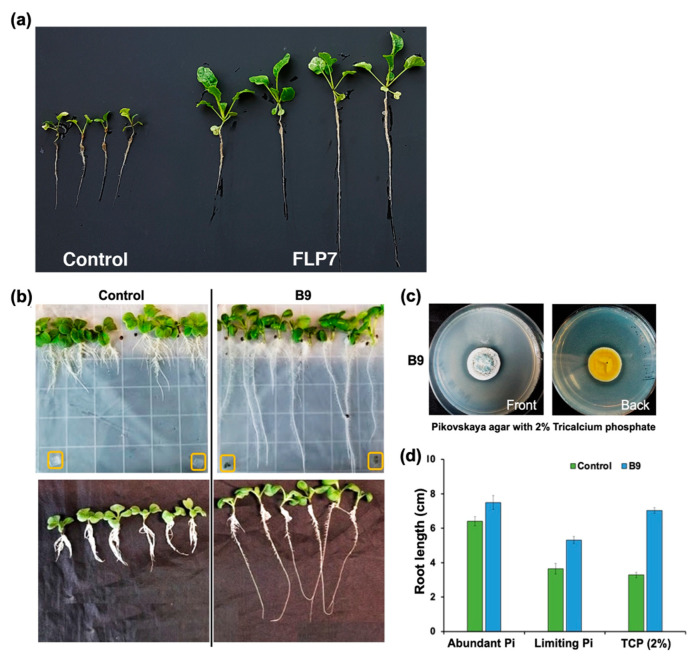
*P. citrinum* enables robust root and shoot growth in Choy Sum under phosphate (Pi)-limiting conditions. (**a**) The morphological traits of Choy Sum treated with FLP7 conidia or with water as control. The average leaf area (cm^2^), root length (cm), the dry weight of roots and aerial parts (mg) determined from both FLP7-treated seedlings and the mock control plants were determined and mentioned in the Section 3. Data represents means ± SDs from 3 replicates consisting of 8 plants in each instance. (**b**) The morphological traits of Choy Sum treated with B9, or with water as control in phosphate-limiting conditions in MS agar (top panels; B9 provided as mycelial plugs (boxed in yellow) or in soil (lower; B9 provided as conidia suspension). Control refers to treatment with water or uninoculated medium plugs (only) in the absence of the fungal mycelia/conidia. The average leaf area (cm^2^), root length (cm), the dry weight (mg) of roots and aerial parts were determined from both B9-treated seedlings and the mock control plants. Data represents means ± SDs from 3 replicates consisting of 16 plants in each instance. (**c**) *P. citrinum* helps assimilate phosphate and enables its bioavailability to the host plants. B9 is capable of solubilizing tricalcium phosphate, as judged by the zone of clearance around edges of the B9 colony cultivated on Pikovskaya agar containing 2% Ca_3_(PO_4_)_2_. (**d**) *P. citrinum* provides solubilized phosphate to host plants to induce root growth in Pi-limiting conditions. Bar graph quantifying average root length in Choy Sum plants grown in the presence of phosphate-replete or phosphate-limiting conditions in the presence or absence of the *P. citrinum* B9 strain.

**Figure 3 jof-09-00420-f003:**
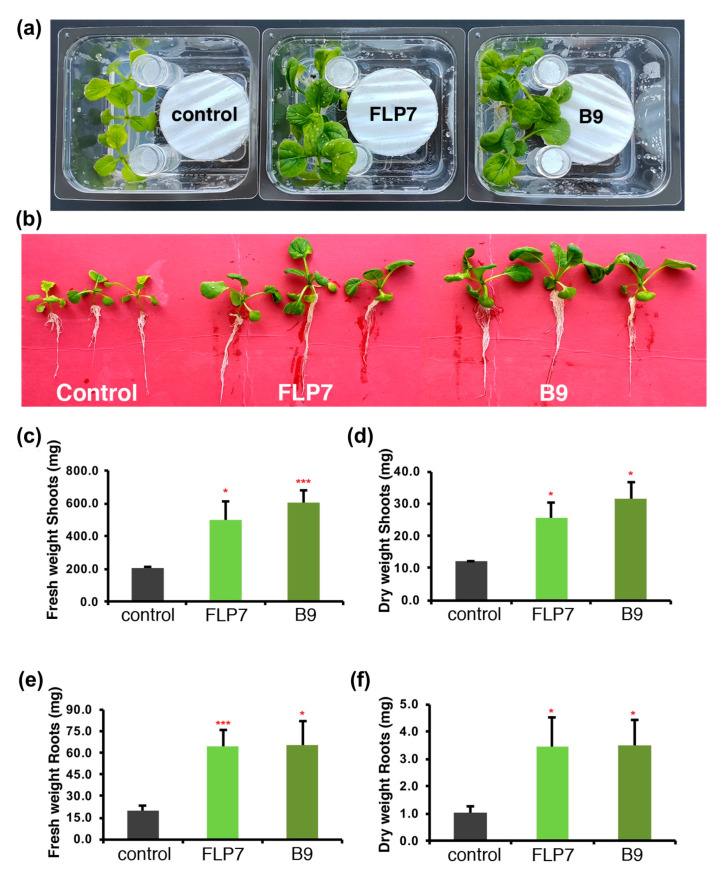
Volatile organic compounds from *P. citrinum* isolates stimulate robust seedling growth in Choy Sum. (**a**) The morphological traits of Choy Sum seedlings grown in tripartite Phytatray II for 10 days; (**b**) the seedlings from (**a**); (**c**–**f**) quantification of the fresh and dry weight of shoots (**c**,**d**) and roots (**e**,**f**) from the seedlings incubated individually with *P. citrinum* B9 or FLP7 conidia. Data represents means ± SDs from 3 replicates each consisting of 8 plants. Barium hydroxide was used to quench excess CO_2_ produced by fungal growth and/or metabolism. Control (mock) inoculation utilized the growth medium (PA, prune agar) without any fungus. Differences were deemed significant at a probability level of *p* < 0.05 (*) or *p* < 0.01(***).

**Figure 4 jof-09-00420-f004:**
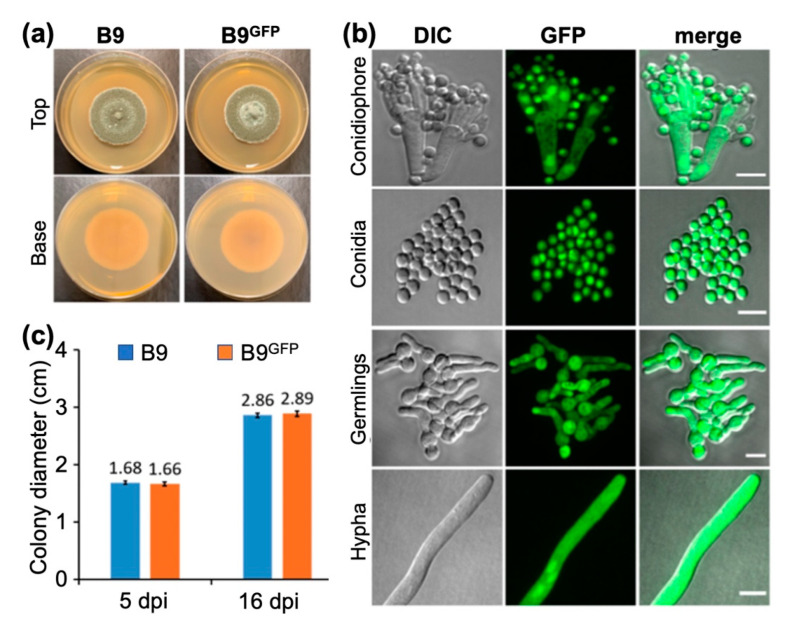
Generation and comparative analysis of the cytosolic GFP-expressing *P. citrinum* isolate B9. (**a**) Colony morphology and growth characteristics of the *P. citrinum* B9 and B9^GFP^ strain. Photographs were taken at 5 dpi (**b**) Confocal microscopy-based confirmation of cytosolic GFP expression in the B9^GFP^ strain at the indicated stages of fungal development. Scale bar equals 5 μm. (**c**) Cytosolic GFP expression does not affect mycelial growth and asexual development or conidiation in the *P. citrinum* B9^GFP^ strain. Conidiation was assessed at day 5 post inoculation, as was growth under constant illumination.

**Figure 5 jof-09-00420-f005:**
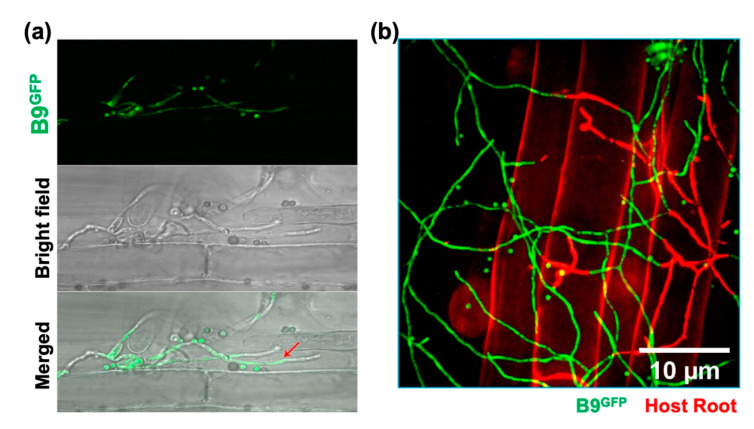
Confocal micrographs of Choy Sum roots incubated with cytosolic eGFP-expressing B9 strain of *P. citrinum*. Choy Sum roots were incubated with 1 × 10^4^ spores/mL at day 5 post germination. The root colonization was analyzed at 1, 2 and 3 days after incubation. Confocal microscopic images of Choy Sum roots incubated with B9^GFP^ strain (**a**,**b**). Root tissue was counter-stained with Propidium iodide. GFP, green fluorescent protein; bright field, host root, Propidium iodide (PI) staining, merge, composite of the GFP, PI and bright field channels.

**Figure 6 jof-09-00420-f006:**
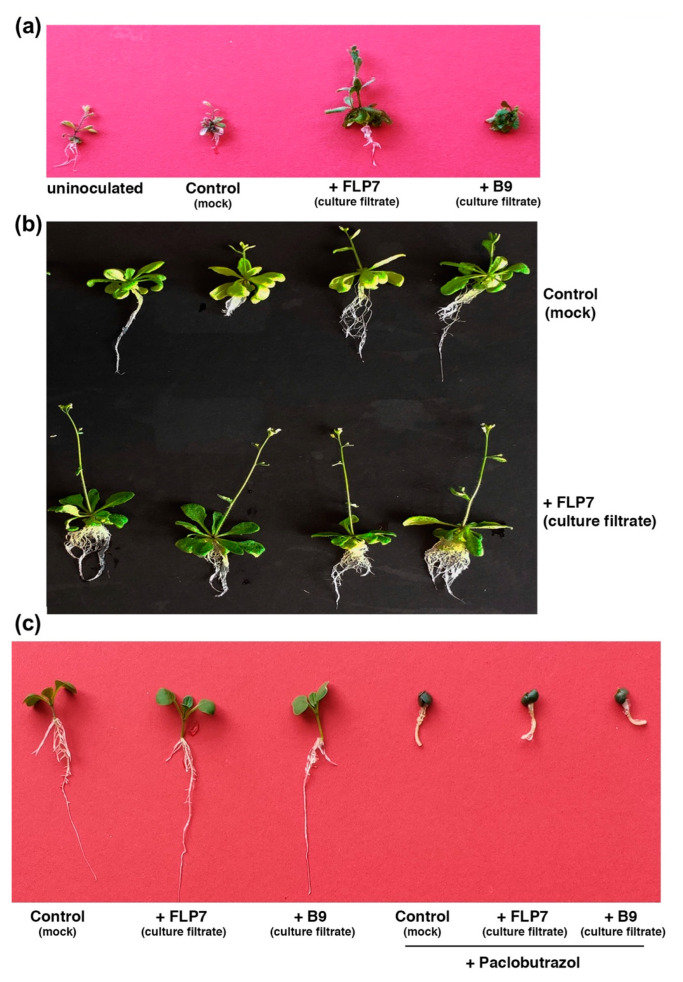
The effect of *P. citrinum* on the growth of Arabidopsis *ga1* mutant and wild-type accession Col-0. (**a**) The morphological traits of Arabidopsis *ga1* mutant in the presence or absence of exudates from beneficial *P. citrinum* cultures (B9 and FLP7) grown in a complete medium. (**b**) Wild-type Arabidopsis Col-0 treated with the culture filtrate from *P. citrinum*. Control refers to the mock inoculation with equivalent amount of growth medium without the fungus. (**c**) Gibberellin(s) produced by *P. citrinum* contribute in part to growth promotion in Choy Sum. The culture filtrate from the indicated *P. citrinum* isolate grown in a complete medium in the presence or absence of Paclobutrazol (the gibberellin biosynthesis inhibitor) was inoculated on Choy Sum seeds. Control refers to mock inoculation with the sterile growth medium in the absence of the fungus.

## Data Availability

All datasets for this study are included in the manuscript and/or the Supplementary Files. All data that support the findings of this study are available from the corresponding authors upon reasonable request.

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
