# Peer review of "Penicillium citrinum Provides Transkingdom Growth Benefits in Choy Sum (Brassica rapa var. parachinensis)"

_jof, 2023, doi:10.3390/jof9040420_

Round 1
Reviewer 1 Report (Previous Reviewer 1)
The document was revised, once it includes the observations made by the reviewers. The wording of all sections was improved. Clear and precise information was included, as well as the corresponding citations.
Author Response
Many thanks for such a positive and encouraging response to our revised/resubmitted manuscript. Thanks for finding the manuscript suitable for publication in Journal of Fungi.
Reviewer 2 Report (New Reviewer)
Dear Authors,
I think you study is of excellent technical quality. I only have a few minor comments, as follows.
Synthesis of a plant phytohormone is not a “mimic” in my opinion, but of course GA1 and GA3 can be parts of a large mixture of non-annotated GA-like acting fungal metabolites in the current setting as well.
L93: what do you mean by “most intricate rhizosphere environment”? Perhaps more prudence is required.
L100: Add citations of PGP studies on Brassicaceae species. There is more than a few.
L162: a bit unclear, please provide a recipe for “complete medium” or revise.
L199: recipe bit unclear, use final in-solution concentrations only.
L248: Was the EtOAc extract evaporated and reuptaken in another solvent (MeOH?) before loading?Please clarify.
L776: Populus should be in italic
Best regards.
Author Response
I think your study is of excellent technical quality. I only have a few minor comments, as follows.
Many thanks for the encouraging remarks, and for finding the overall technical quality to be excellent.
(1) Synthesis of a plant phytohormone is not a “mimic” in my opinion, but of course GA1 and GA3 can be parts of a large mixture of non-annotated GA-like acting fungal metabolites in the current setting as well.
We have now changed the word “mimic” and included a context dependent change to the description of the fungus-derived phytohormones. Referring to them as either fungus-derived GAs (Line 479) or P. citrinum-derived phytohormones and/or derivatives (Line 506) or mycobiota-derived phytohormones and/or derivatives (Line 552). Lastly, the Sub-heading for this Results section 3.5 now reads: P. citrinum produces the phytohormones Gibberellin, and Cytokinin.
(2) L93: what do you mean by “most intricate rhizosphere environment”? Perhaps more prudence is required.
Many thanks for pointing this out. This sentence has now been toned down to: However, the microbiome that shapes the rhizosphere environment of Choy Sum remains largely unexplored.
(3) L100: Add citations of PGP studies on Brassicaceae species. There is more than a few.
References 6-10 have now been added to cite the requisite PGP studies in Brassicaceae species (Line 67).
(4) L162: a bit unclear, please provide a recipe for “complete medium” or revise.
The composition of the Complete Medium has now been included in the revised manuscript (Line 113).
(5) L199: recipe bit unclear, use final in-solution concentrations only.
Many thanks. We have now included “standard induction medium” as a descriptor, and provided the actual concentrations of all the components therein. This now reads: standard induction medium [K salts (10 mM K2HPO4, 10 mM KH2PO4); M salts (2 mM MgSO4-7H2O, 2.5 mM NaCl, 4 mM NH4NO3, 0.7 mM CaCl2, 10 mM FeSO4); with glucose 5 mM, MES 40 mM, glycerol 0.5%, 100 μg/mL Kanamycin, 200 μg/mL acetosyringone].
(6) L248: Was the EtOAc extract evaporated and reuptaken in another solvent (MeOH?) before loading? Please clarify.
This has now been clarified as follows: Line 201-207: The supernatant of the mixture was then transferred and evaporated to dryness. For SPE purification, samples were reconstituted in 100% methanol and loaded onto the C-18 cartridge preconditioned with 6 mL methanol and equilibration with 6 mL distilled water. The cartridges were washed with 5 mL distilled water and the retained phytohormones (GAs) were eluted with 6 mL 100% methanol containing 1% (v/v) formic acid. The eluted extract was evaporated to dryness and reconstituted in 50% (v/v) methanol for the following LC-MS/MS analysis.
(7) L776: Populus should be in italic.
Populus has now been italicised. The generic/Latin names in Bibliography have been italicised too.
We thank the Reviewers for their critique and suggestions, and for their help in improving the manuscript further.
This manuscript is a resubmission of an earlier submission. The following is a list of the peer review reports and author responses from that submission.
Round 1
Reviewer 1 Report
The manuscript is very interesting and the results it presents are significant. I consider that it has an adequate writing and a robust data analysis. For this reason, I only recommend correcting details such as the italics of scientific names and improving the writing of paragraphs.
Reviewer 2 Report
Reviewed paper deal with growth promotion effects of Penicillium citrinum on broadly cultivated crops. From this point of the view, this kind of studies are always wanted and desired. However, effects of the Penicillium citrinum in promoting of plants growth are well known, thus manuscript does not bring any new information or conclusion.
Overall, I found several major points which should be addressed
1. overal design of the study seems to be quite chaotic. Title address intention to study effects of the penicilium citrinum on plant growth promotion. Abstract explains, that root diversity was screened and P. citrinum was found somehow randomly. Overal design of the methods again showed intention to study P. citrinovirens on purpose, and finally, first chapter of the Results "3.1 Isolation of Beneficial Fungi That Enhance Plant Growth " indicate that aim of the study was to assess overal diversity of the plant´s roots.
2. Study completely lacks relevant literature. In total 21 references do not cover state-of-art and is exceptionally low in this type of studies. Authors ommitted literature relevant to P. citrinum effects. In the Introduction, there is no mention about well-known effects of P. citrinum or known production of giberellins and other volatile compounds. It gives an impression, that those informations were omitted on purpose in order to make presented results more unique. Introduction has to be completely re-written with stronger focus on studied species.
3. In the chapter Results and discussion, there is completely missing discussion part. Authors should put their results to the context of already published informations. Instead of that, they use strong sentences like Interestingly, we detected high amount of gibberelins (abstract) despite the fact, that production of gibberellins by P. citrinum is well-known. This gives an impression, that authors lacks better insight into topic.
4. Paper seems to be written in rush. Multiple formatting errors are appearing such as not properly cited references (line 188), formatting of chapter and subchapter titles (once initiall letter capital, once not), latin name in italics (e.g. line 430) etc. Therefore, I strongly suggest deep revision of the manuscript.